# Development and Characterization of Phosphate Glass Fibers and Their Application in the Reinforcement of Polyester Matrix Composites

**DOI:** 10.3390/ma15217601

**Published:** 2022-10-29

**Authors:** Nezha Saloumi, Iliass Daki, Mehdi El Bouchti, Mina Oumam, Bouchaib Manoun, Mohamed Yousfi, Hassan Hannache, Omar Cherkaoui

**Affiliations:** 1Textile Materials Research Laboratory (REMTEX), Higher School of Textile and Clothing Industries (ESITH), Casablanca 20000, Morocco; 2Engineering and Materials Laboratory (LIMAT), Faculty of Science Ben M’Sik, Hassan II University, Casablanca 20670, Morocco; 3Materials Science and Nanoengineering Department, Mohamed VI Polytechnic University, Benguerir 43150, Morocco; 4Radiation-Matter and Instrumentation, Hassan First University of Settat, FST, Settat 26000, Morocco; 5Univ Lyon, CNRS, UMR 5223, Polymeric Materials Engineering, University Claude Bernard Lyon 1, INSA Lyon, University Jean Monnet, F-69621 Villeurbanne, France

**Keywords:** phosphate glass fibers (PGF), polyester matrix composites, chemical durability, mechanical properties, morphological properties

## Abstract

This study focused on the production and characterization of phosphate glass fibers (PGF) for application as composite reinforcement. Phosphate glasses belonging to the system 52P_2_O_5_24CaO13MgO (11-(X + Y)) K_2_OXFe_2_O_3_YTiO_2_ (X:1, 3, 5) and (Y:0.5, 1) were elaborated and converted to phosphate glass fibers. First, their mechanical properties and chemical durability were investigated. Then, the optimized PGF compositions were used afterward as reinforcement for thermosetting composite materials. Polyester matrices reinforced with short phosphate glass fibers (sPGF) up to 20 wt % were manufactured by the contact molding process. The mechanical and morphological properties of different sPGF-reinforced polyester systems were evaluated. The choice between the different phosphate-based glass syntheses (PGFs) was determined by their superior mechanical performance, their interesting chemical durability, and their high level of dispersion in the polyester matrix without any ad sizing as proven by SEM morphological analysis. Moreover, the characterization of mechanical properties revealed that the tensile and flexural moduli of the developed polyester-based composites were improved by increasing the sPGF content in the polymer matrix in perfect agreement with Takayanagi model predictions. The present work thus highlights some promising results to obtain high-quality phosphate glass fiber-reinforced polyester parts which can be transposed to other thermosetting or thermoplastic-based composites for high-value applications.

## 1. Introduction

The worldwide consumption of composite materials, namely fiber-reinforced polymers, is increasing due to their high use in the automotive, aerospace, construction, and wind energy sectors [1,2]. This wide use is due to their excellent insulating properties and high resistance to fatigue and corrosion [3,4]. Currently, the main fibers used to manufacture organic matrix composites consist of glass fibers (GF) and carbon fibers (CB). Today, these polymer-based composites have many structural applications in different fields: nautical, automotive, aeronautical, and biomedical [5]. Since the beginning of the development of polymer composites, the major problem of thermosetting matrix-based composites is the type of reinforcements used in terms of cost and their compatibility (physical and chemical) with the matrix and also composition, mechanical and chemical properties, and their adequate surface energy. In this regard, researchers have put significant effort to improve their properties according to customer needs and applications. Polyester resin is extensively used as a matrix for composites in many industries, such as transportation, electrical, and construction, due to their good corrosion resistance, low density, simple processing, low cost, and strong compression and shear resistances [6]. Phosphate glass fibers have attracted much attention for use as reinforcing agents in composite materials and nowadays are considered a potential substitute for commercial silica-based glass fibers [7]. The glass fibers are strong in tensile and weak in compression and shear. When combined with a polymer resin, these materials form structural composite parts that were strong in compression, tension, and bending [8]. However, their higher price is the main factor limiting the engineering application. In contrast, phosphate glass fibers have a lower price and better toughness, which is popular in some composite applications. In this regard, phosphate glass fiber-reinforced polyester composites are lightweight and less expensive than carbon fiber and silica glass composites. The low chemical durability of phosphate glass fibers limits their utilization as reinforcement material. Therefore, many efforts have been made to increase their durability. Hence, it would be beneficial to produce glass fibers having both good chemical durability and physical properties identical to those of competing silica-based fibers [9]. The structure of pure phosphate glasses is constituted of [PO_4_] tetrahedral which has three shared corners connected by bridges of oxygen, or (BO)s [10,11]. Several studies have been made to improve the chemical durability of phosphate glasses, among the results established is the addition of metal oxides in the glass network which results in the breaking of the P-O-P bond and the formation of a P-O-M bond which is more resistant. Therefore, the non-bridges oxygen (NBO)s, are created in the glass network and the orthophosphate (Q^0^) is transformed into chains of pyrophosphate (Q^1^) and the chains become smaller and smaller until metaphosphate (Q^2^) and three-dimensional (Q^3^) anions are reached [12,13]. Among these oxides are iron and titanium oxides. Titanium dioxide TiO_2_ is an intermediate glass-forming oxide that is very useful due to its chemical stabilization and degradability control of phosphate glasses (PG). Therefore, several studies have proved that TiO_2_ metal oxide is effective for the enhancement of PG physical properties such as optical, magnetic, and electrical properties [14,15]. It has been found that the titanium ions penetrate the phosphate glass network as [TiO_6_], [TiO_5_], or [TiO_4_] elements [16]. These elements result in depolymerization where the P-O-P bonds are replaced by the Ti-O-P bonds which are more resistant [17]. On the other hand, many studies have been carried out on the effect of iron oxide on phosphate glass systems [18,19,20]. It has been shown that the addition of iron oxide into the phosphate glass composition improves their chemical durability. This effect is due to the replacement of P-O-P bonds by Fe-O-P bonds and reinforcement of the cross-linking of phosphate chains. Furthermore, this stronger ionic cross-linking could also reduce the crystallization tendency of phosphate glass and increase its mechanical properties [21].

The aim of the current study is the manufacturing of phosphate glass fibers (PGF) with high mechanical and chemical performance and their use as reinforcement of polyester matrix composite materials. Therefore, the study of improvement of the chemical durability and mechanical properties of the PGF fillers is studied by doping the composition of phosphate glass fibers with Fe_2_O_3_/TiO_2_ elements. The most performing formulation of phosphate glass fibers obtained is used as a reinforcement of composite materials. On the other hand, polyester composites reinforced by short phosphate glass fibers (sPGF) are produced and evaluated to determine their mechanical and morphological properties.

## 2. Materials and Methods

### 2.1. Synthesis and Characterization of Phosphate Glass Fibers

Phosphate glasses were prepared by the direct melting method from suitable raw material mixtures: potassium hydrogen phosphate K_2_HPO_4_, tri-calcium phosphate Ca_3_(PO_4_)_2_, magnesium phosphate (Mg(H_2_PO_4_)_2_), iron phosphate (FePO_4_), and titanium oxide (TiO_2_) in suitable proportions (Table 1). The mixtures were placed in crucibles in the electrical furnace (Carbolite, LE MANS, France) at a temperature of 300 °C for 1 h to remove H_2_O and CO_2_, and then the temperature was slowly increased to 1000 °C with a heating rate of 6 °C/min. The molten glasses were kept for about an hour at this high temperature and then poured into a stainless-steel tubular mold to obtain batches of glass weighing about 6 g. The elaborated phosphate glasses were transformed into fibers using the Melt-drawing spinning process. In this case, 70 g of the phosphate glass was placed into the furnace of the spinning machine for 1 h to melt the glass completely. The spinning temperature was programmed at 600 °C, which is above glass transition temperatures. Once, the glass has become homogeneous liquid was pulled and collected (Figure 1). The Density measurements of prepared fibers were made by the method of Archimedes. The structural properties of fibers were studied by FTIR analysis using a Bruker TENSOR 27 FTIR spectrometer (Bruker Corporation, Markham, ON, Canada) in this case, pellet samples were prepared by pressing a mixture of phosphate glass powder (1%) and anhydrous KBr powder (99%). The chemical durability of elaborate phosphate glass fibers was evaluated using the weight loss method in different pH aqueous mediums. However, 300 mg of phosphate glass fibers of length 20 mm were introduced into glass vials containing 30 mL of solutions (distilled water, buffer solutions (pH = 4.5, 5.5, and 8.5)). In this case, all vials containing phosphate glass fibers were placed into a heated bath at 37 °C for 360 h. The fibers removed from these solutions were placed into the oven at 25 °C for 24 h before the final weight measurement. The morphology of fibers was analyzed by SEM using the instrument JEOL JCM 6000Plus (Therm Fisher Scientific, Waltham, MA, USA) coupled to an energy dispersive X-ray analysis (SEM-EDX, model FEI Type Quattro S, Therm Fisher Scientific, Waltham, USA). The mechanical properties were determined by a single filament tensile test based on the standard ISO 11566. A single fiber (approximately 11 μm) was primarily mounted to the paper frame with a gauge length of 25 mm. The fiber was bonded to the frame with an adhesive Universal Bison (Universal Bison KIT, Rotterdam, The Netherlands). For each phosphate glass fiber composition, thirty samples were prepared and tested using a universal testing machine Ludwig MPK equipment (Mpk LUDWIG UG, Emsbüren, Germany), with clamps attached to a 1 N sensor, and tested at room temperature with a displacement rate of 1 mm min^−1^. The best-obtained composition of phosphate glass fiber which is the most chemically stable in different aqueous media and has the most interesting mechanical properties was used in the elaboration of polyester composite materials in the following sections.

In this study, all fibers were produced at a speed of 500 m/min and subsequently analyzed by an optical microscope to determine the diameter of all developed phosphate glass fibers. Thirty samples were analyzed and showed that the produced phosphate glass fibers produced have a diameter of (11 ± 0.5) μm on average.

### 2.2. Preparation and Characterization of Polyester Composites Materials

#### 2.2.1. Preparation of Polyester Resin

Polyester resin is a thermosetting matrix that is widely used in plastic composites because of its low cost, chemical resistance, and rapid drying process. The density of the polyester resin used is 1200 kg/m^3^ and its viscosity at 25 °C is 640 mPa·s. The drying of polyester resin requires the addition of an accelerator (cobalt) and a catalyst (methyl ethyl ketone peroxide) respectively. The polyester resin, cobalt, and methyl ethyl ketone peroxide were obtained from detail Chimie (Casablanca, Morocco). In the present study, we used a resin formulation with 2% methyl ethyl ketone peroxide and 0.24% cobalt accelerator.

#### 2.2.2. Elaboration of Composites Materials

Phosphate glass fiber-reinforced polyester composites were prepared by the contact molding technique at a temperature of 25 °C. The fiber content varied from 5 to 20 wt %. In an open bowl, the polyester resin was thoroughly mixed with 2% of methyl ethyl ketone peroxide and 0.24% of cobalt octoate. This prepared resin was mixed with phosphate glass fibers with the system 52P_2_O_5_-24CaO-5K_2_O-13MgO-5Fe_2_O_3_-1TiO_2_ and the fiber length was 3 ± 1 mm. Here, the mixtures were mechanically stirred at 900 rpm/min for 2 min, then poured into an aluminum mold prepared according to dimensions determined by ISO 527 and ISO 14125 and treated with a release wax. After 2 h, the resin is cured and the composite part was obtained.

#### 2.2.3. Characterization of Composites Materials

The properties of all developed composites are studied. However, their physical characteristics such as mass fraction, volume fraction, thickness, and density were determined. In addition, their mechanical properties were studied by the tensile and the Flexural testes. The tensile and flexural tests were carried out according to ISO 527 and ISO14125 using a Zwickel universal testing machine. three samples were tested for each composite with a speed of 2 mm/min at ambient temperature. The morphological properties of the manufactured composites were also evaluated by SEM analysis using a JEOL JCM 6000Plus type scanning electron microscope at an accelerating voltage of 15 kV. And, the study of the mechanical modeling of composites is carried out by the Takayanagi model.

## 3. Results and Discussion

### 3.1. Characterization of Phosphate Glass Fibers

#### 3.1.1. Infrared Spectroscopy Analysis

Figure 2 shows the infrared analysis spectra of the elaborated phosphate glasses. The results of this study show the existence of four intense bands observed at ~520, ~750, ~1070, ~1300 cm^−1^, and a large intense band between ~850 and ~1200 cm^−1^. The band observed at ~520 cm^−1^ is attributed to the harmonics O=P-O bond of (PO_4_)^3–^ [22]. Asymmetrical stretching of bridging oxygen (P-O-P) vsy in the Q^2^ species was observed around ~750 cm^−1^ [23]. Compared to the reference phosphate glass formulation, the intensity of the bands 520 and 750 cm^-1^, decreased with the addition of 1, 3, and 5 mol % Fe_2_O_3_, and 0.5 mol % TiO_2_ to the glass composition. This result has also been proven by Chao Tan [24]. The band around 1095 cm^−1^ is attributed to the asymmetric stretching (PO_3_) _vasy_ characteristic of the Q^1^ groups. Finally, the bands around ~850 and 1200 cm^−1^ were associated with asymmetric stretching vibrations (P-O-P) _vasy_ in Q^2^ and Q^1^ tetrahedral, respectively which is in agreement with the results of Valappil [25]. A decrease in intensity was observed by the combination of the Fe_2_O_3_ and TiO_2_ elements in the phosphate glass composition.

#### 3.1.2. Chemical Durability

The chemical durability study was carried out by the mass loss method of phosphate glass fibers in aqueous media at different pH (8.5, 6.5, 5.5, and 4.5) at a temperature of 37 °C. Figure 3 shows the results of this study as a function of time. As can be seen, the substitution of K_2_O with 0.5% and 1%mol of TiO_2_ and (1%, 3%, 5%mol) of Fe_2_O_3_ in the P52Fe0Ti0 glass composition decreased the percentage of weight loss from 100% to 6.66% at pH = 6.5, from 96% to 5.3% at pH = 4.5, 98.5% to 5.2% at pH = 5.5, and from 90% to 4% at pH = 8.5. This effect is due to the release of Ti^4+^ and Fe^3+^ ions that improve the cross-linking of phosphate units compared to Ca^2+^, Mg^2+^, and K^+^. However, doping of the phosphate glass composition with Fe_2_O_3_ and TiO_2_ oxides causes chemical stability of the phosphate glass fibers in aqueous media at different pH. This effect may be due to the penetration of Fe^3+^ and Ti^4+^ ions in the glass network, which causes cleavage of the P-O-P bonds and the creation of Fe-O-P and Ti-O-P bonds, which are more compact and resistant to hydration than the P-O-P bond [26]. This interruption reduces the species Q^1^ quantity compared to the species Q^2^ [27,28]. Therefore, as observed in Figure 3, with the increase of the percentage of these elements in the glass composition, the percentage of fiber loss decreases with the change in the conditions, which confirms the chemical stability of the Fe_2_O_3_/TiO_2_ doped phosphate glass fibers.

In addition, the increase in chemical durability of phosphate glass fibers by the addition of these elements in the glass network depends on the decrease in the hydration energy of (Q^2^ > Q^1^ > Q^0^), which increases the hydrolytic stability of phosphate units [29]. The results obtained in this study were compared with those found by Ensanya Ali Abou Neel [30], Ray [31], and Marasinghe [32] and show that the chemical durability of the developed phosphate glass fibers is more important, which proves the high chemical performance of these fibers.

Table 2 shows the degradation rates of the elaborated phosphate glass fibers. In the first step, we set the TiO_2_ content at 0.5% and varied the mol % of Fe_2_O_3_ by 1%, 3%, and 5%, and then we set the content of TiO_2_ at 1% and varied the mol % of Fe_2_O_3_ by 1%, 3%, and 5%. The substitution of K_2_O by the combination of Fe_2_O_3_ and TiO_2_ in the same glass composition has shown a significant enhancement of the degradation rate of phosphate glass fibers. It can be concluded that the combination of 1% of TiO_2_ and 5% of Fe_2_O_3_ presents a better solution to produce a stable phosphate glass fiber in both alkaline and acid environments.

#### 3.1.3. Fibre Surface Morphology

Scanning electron microscopy analysis of the phosphate glass fibers before and after the degradation tests at 37 °C is shown in Figure 4. The results confirm the smooth morphology of the recently produced phosphate glass fibers as shown in Figure 4a. After 3 days of degradation, we noticed a degradation of the surface layer of the phosphate fiber which is very noticeable in the formulation of phosphate glass fibers without Fe_2_O_3_ and TiO_2_ elements (Figure 4b). In contrast, the phosphate glass fibers compositions containing Fe_2_O_3_ and TiO_2_ are much more resistant compared to the reference formulation P52Fe0Ti0 up to 20 days (Figure 4c,d).

#### 3.1.4. Single Filament Tensile Test

Figure 5 shows the mechanical properties of phosphate glass fibers produced by tensile tests based on ISO11566. The results indicate that the tensile strength value of phosphate glass fibers increases from (0.85 ± 0.06) GPa to (1.18 ± 0.08) GPa and (1.5 ± 0.1) GPa and the tensile modulus increase from (53 ± 1) GPa to (63 ± 1) GPa and (79.52 ± 1) GPa with the addition of 0.5% and 1%mol TiO_2_ in the glass compositions P52Fe0Ti0.5 and P52Fe0Ti1. Furthermore, the addition of 1%, 3%, and 5%mol Fe_2_O_3_ in the formulations of glass fibers P52Fe1Ti0, P52Fe3Ti0, and P52Fe5Ti0 led to tensile strength values of (1.61 ± 0.08), (1.83 ± 0.08) and (2.13 ± 0.1) GPa, and to tensile modulus values (85 ± 2) GPa to (94 ± 2) GPa and (109.2 ± 2) GPa. Figure 5 also shows that the tensile properties increase with the combination of Fe_2_O_3_ and TiO_2_ elements in the phosphate glass fiber compositions over the formulations of P52Fe0Ti0, P52Fe1Ti0, P52Fe3Ti0, P52Fe5Ti0, P52Fe0Ti0.5, and P52Fe3Ti0 fibers. The obtained tensile strength values are (1.33 ± 0.09), (1.56 ± 0.05) and (1.86 ± 0.08) GPa, and the tensile modulus values are (69.2 ± 1) GPa, (64.2 ± 1) GPa and (95.2 ± 2) GPa for P52Fe1Ti0.5, P52Fe3Ti0.5, and P52Fe5Ti0.5 glass formulations. In addition, the tensile strength values increase from the last values obtained when TiO_2_ is fixed at 1%mol and Fe_2_O_3_ varies from 1 to 5%mol. The tensile strength values obtained were (1.72 ± 0.07), (1.89 ± 0.09), and (2.23 ± 0.1) GPa and modulus values are (87.79 ± 1) GPa, (96.68 ± 2) GPa, and (112.0 ± 2) GPa for the reference formulations P52Fe1Ti1, P52Fe3Ti1, and P52Fe5Ti1. A comparative study on the tensile strength and moduli of glass fibers was performed to compare the obtained results with those found in the literature. The developed phosphate glass fibers in this study exhibit physical properties meeting or exceeding the mechanical properties of conventional glass fibers [21,33,34,35].

After the evaluation of the properties of the developed phosphate glass fiber, we found that the addition of Fe_2_O_3_/TiO_2_ in the composition of phosphate glasses improves their mechanical properties and chemical durability. This improvement is due to the creation of Fe-O-P and Ti-O-P bonds in the glass network which is compact and resistant. The 52P_2_O_5_-24CaO-5K_2_O-13MgO-1TiO_2_-5Fe_2_O_3_ glass fiber system presents the most important properties among the developed series of fibers. This fiber is chemically stable in different aqueous media with a tensile strength of 2.23 GPa and a Young modulus of 112 GPa. These results allow us to apply this fiber in the development of polyester matrix composites.

### 3.2. Characterization of Phosphate Glass Fiber-Reinforced Polyester Composites

#### 3.2.1. Physical Characteristics of Composites

The mass and volume fractions of fibers in the composite depend on the reinforcement used, the reinforcement/resin mixture, and the existence of voids. These parameters have a strong influence on the properties of the composites. The good adhesion between the matrix and the reinforcement increases the mechanical properties of the composites [36]. In the present work, the average density of the used reinforcement (PGF) is 2.7 g·cm^−3^ and the density of the cured polyester resin is 1.2 g·cm^−3^. The physical characteristics of the developed composites were determined and presented in Table 3.

The volume content of phosphate glass fibers $\emptyset {\left(\mathrm{PGF}\right)}_{\mathrm{vol}}$ is calculated from composite density measurements using the following formula:(1)$$\emptyset {\left(\mathrm{PGF}\right)}_{\mathrm{vol}}=\frac{m_{\mathrm{PGF}}}{m_{t}}\;\cdot \;\frac{\rho_{c}}{\rho_{\mathrm{PGF}}}$$
where $m_{\mathrm{PGF}}$ is the mass of fibers (g), $m_{t}$ is the total mass of fibers plus resin, $\rho_{c}$ is the composite density and $\rho_{\mathrm{PGF}}$ is the fiber density.

From these results, we can observe that the density value of the elaborated composites increases from 1.24 to 1.36 g·cm^−3^ with increasing the fiber content in the polyester resin from 5 wt % to 20 wt % respectively. This increase is due to the high-density value of the phosphate glass fibers (2.7 g·cm^−3^), and the complete wetting of the PGF in the polyester resin, which results in the filling of the void inside the composites [37,38]. This assumption will be further confirmed in the next sections.

#### 3.2.2. Morphological Properties of Composites

SEM micrographs of the fractured surfaces of the composites were captured to visualize the fiber/matrix adhesion in phosphate glass fiber-reinforced polyester composites. Figure 6 shows the SEM pictures of the polyester matrix and the reinforcement (phosphate glass fiber). It can be seen the presence of a smooth surface and uniform diameter (11 μm) of the phosphate fibers used (Figure 6a). Furthermore, the fractured surface of the cured polyester resin is shown in Figure 6b, which confirms that the matrix surface is smooth, uniform, and without imperfections.

The micrographs of the fractured surfaces of the different glass fiber-reinforced polyester composites (5%, 12%, 16%, and 20% in weight fractions) were presented in Figure 7. The PGF fibers were well dispersed in the polyester matrix due to the compatibility of the phosphate glass fibers with the polyester matrix. Erden [39] studied the morphology of glass fiber-polyester composites using SEM and found a fiber pull-out failure phenomenon in the case of glass fiber-reinforced polyester composite. moreover, they observed a very small amount of polyester resin adhered and wetted on the surfaces of glass fibers in the fracture zone of the composite, and the presence of glass agglomerates and microvoids in the composites indicating the poor interfacial bonding between the polyester matrix and the fibers. In contrast, PGF-reinforced composites developed in this work have a high mechanical behavior. This effect is due to the nature and outstanding mechanical properties of the phosphate glass-based reinforcement which limits the internal cracking of the polyester matrix.

#### 3.2.3. Mechanical Properties of Composite

##### Tensile Testing

Figure 8 illustrates the effect of fiber content on the tensile strength of phosphate glass fiber-reinforced polyester composites. The addition of the phosphate fibers improved the tensile strength values of the polyester matrix significantly. The tensile strengths of the polyester composites increased from (11 ± 1) MPa to (22.12 ± 2) MPa with increasing fiber content from 0%vol to 4.1%vol and achieved its maximum value of (43.75 ± 2) MPa with increasing fiber content to 9.2%vol. The addition of phosphate glass fibers to the polyester resin limits the propagation of cracks in the composites. Thus, the strength of the PGF-reinforced polyester increases with the increase in the content in the fibers, which improves their ductility [40]. This significant improvement in the tensile strength of the polyester matrix is due to the high mechanical performances of the phosphate glass fibers used which have a high strength of 2.23 GPa. The reinforcing effect of phosphate glass fibers is attributed to the strong interfacial interaction between the polyester matrix and the glass fibers. Figure 8 also shows that the tensile modulus of the processed composites increased with increasing fiber content. With 0%vol, 2.3%vol, and 9.2%vol of fibers, the tensile modulus was 0.92 GPa, 1.1 GPa, and 2.3 GPa, respectively. This improvement in modulus and tensile strength of the composites developed is attributed to the mechanical strength of the PGF used which causes void filling in the polyester matrix.

##### Flexural Testing

The bending properties of the developed composite materials were presented in Figure 9. The results show that the flexural strength and modulus were improved by increasing the phosphate glass fiber content in the polyester composites. The Polyester/20 wt %PGF composite represents the maximum value of flexural strength and flexural modulus, which attain 45 MPa and 2.93 GPa, respectively, and represent a 24.2% improvement compared to the polyester matrix without phosphate glass fibers. This improvement is attributed on the one hand to the strong interfacial interaction between the polyester matrix and the glass fibers and on the other hand to the high strength of the phosphate glass fibers used.

A literature review was carried out and a comparison of the results related to the mechanical properties of the composites obtained with some polyester matrix composites reinforced by silica glass fibers showed that the results obtained in this study are in agreement with the literature [41,42]. The results confirmed that the phosphate glass fibers developed in this work are very competitive with the silica-based glass fiber found in the market and could be used as a promising reinforcement in the field of composite materials.

In this study, the choice among the different synthesized phosphate-based glass (PGFs) was dictated by their superior mechanical performance, their interesting chemical durability, and their high level of dispersion in the polyester matrix as proven by SEM measurements. Suggesting and validating some relevant mechanisms related to the interface mechanical performances of fiber/resin requires the use of a combination of additional methods of analysis such as wetting measurements (surface energy of the fibers and the resin and contact angle between the filament and the resin droplet for each phosphate glass fiber composition), interfacial shear strength, IFSS, from micro debonding test and interlaminar shear strength, ILSS, from short-beam shear test [43]. For example, in the fiber pull-out technique, the failure is at the fiber-matrix interface. Thus, a debonding force or stress can be determined allowing us to assess the interfacial shear strength. In addition, measurements of work of fracture and friction characteristics of the interface between glass fibers and polyester resins could be determined. Because of this, it is possible to examine the types of failure encountered in short glass fiber-reinforced polyester or epoxy composites [44,45]. These complementary experiments are now in progress and will be the subject of a featuring article.

##### Mechanical Modeling

A Takayanagi model was used to approximate the moduli values of the composites taking into account the interface contribution. The Takayanagi model considers the material as composed of two elements working in parallel. The first element consists of the entire reinforcing fiber and part of the polymer matrix, acting in series. On the other hand, the rest of the matrix represents the second element [46]. The expression of the composite modulus using the Takayanagi prediction is:(2)$$E_{c}{=E}_{m}\left(1-\frac{V_{f}}{\Phi }\right)+\left(\frac{V_{f}{\;E}_{m}{\;E}_{f}}{E_{m}\Phi^{2}+\left(1-\Phi \right){\Phi \;E}_{f}}\right)$$
where E_f_ and E_m_ are the moduli of the fiber and the matrix, E_c_ is the modulus of the composite, and Φ is an adjustable parameter. The parameter Φ could be related to the morphology of the composite.

According to different literature reports [47], the series/parallel-based Takayanagi model (a “two-phase” model) is very suitable and widely used in case of binary systems such as particulate-filled amorphous polymers and fiber-reinforced thermosetting composites. It predicts with high accuracy the elastic moduli of filler-reinforced composites by considering matrix/filler interaction. In addition, the Takayanagi model was chosen in this study among other micromechanics approaches because it agrees well with our experimental data which allows for examining qualitatively the level of adhesion at the interface of phosphate glass fiber/polyester composites.

In this study, a comparison between the experimental results and the theoretical predictions (Figure 10) gives a mean value of the parameter Φ of 0.97 and 0.98 in tensile and flexural experiments respectively indicating a higher efficiency of stress transfer between phosphate glass fibers and polyester matrix and corroborating the SEM analyses results discussed above.

In Takayanagi experiments, the elastic properties of composites (i.e., tensile young’s moduli) were performed at an initial strain rate ε less than 7 · 10^−3^ s^−1^ [48]. In this study, tensile tests on composites were performed at a constant cross-head speed of 2 mm/min, which corresponds to an initial strain rate of 4 · 10^−4^ s^−1^. However, in the case of filler-reinforced composites with Tg higher than 25 °C, the deformation of the material is rather not strain-rate sensitive at small ε. McClung [49] showed that the variation of the strain rate from 10^−4^ to 10^−2^ s^−1^ didn’t affect the measured tensile moduli of epoxy-based materials since their glass transition Tg was above the ambient testing temperature. Similar behavior was observed by Richeton [50] in the case of amorphous polymers (PMMA and PC). The authors observed a strain rate effect on young’s modulus starting from an ε of 1 s^−1^. Therefore, a strain level up to 1 s^−1^ could correspond to the parameter Φ of value 0.97 to 0.98.

## 4. Conclusions

This paper has presented the production of Fe_2_O_3_/TiO_2_ doped phosphate glass fibers. The mechanical properties and chemical durability of these fibers were studied to determine the best composition that can be used in the reinforcement of composite applications. We found that the addition of Fe_2_O_3_/TiO_2_ in the composition of the glasses increased the chemical durability and tensile strength of the phosphate glass fibers. The best composition of the fibers obtained from this study was 52P_2_O_5_-24CaO-5K_2_O-13MgO-1TiO_2_-5Fe_2_O_3_. The study of the properties of this composition shows their high chemical stability in different aqueous media (acidic and basic) with a tensile strength of 2.23 GPa. This phosphate glass fiber is used as a reinforcement of composite materials. However, polyester matrix composites reinforced by short phosphate glass fiber with a weight fraction varied from 5% to 20% have been elaborated by the contact molding technique. The physical, mechanical, and morphological properties of these composites have been studied. The mechanical analysis of the developed composites showed that the tensile and flexural strength was increased from 11 to 43.8 MPa and from 10.9 to 45 MPa, respectively when 20 wt % PGF was used. In addition, the analysis of morphological properties shows good interfacial adhesion between polyester resin and phosphate glass fiber. Finally, the composites moduli were predicted from the Takayanagi model including a parameter Φ which reflects the level of stress transmission at the matrix/fiber interface. The calculated parameter Φ was close to unity indicating a higher efficiency of stress transfer between PGFs and the polyester matrix.

Hence, the present study highlights the feasibility of using phosphate glass fibers as effective fibrous reinforcements in polymer matrices with good interfacial and mechanical properties by optimizing the composition during the elaboration step of the PGFs and without the addition of sizing.

## Figures and Tables

**Figure 1 materials-15-07601-f001:**
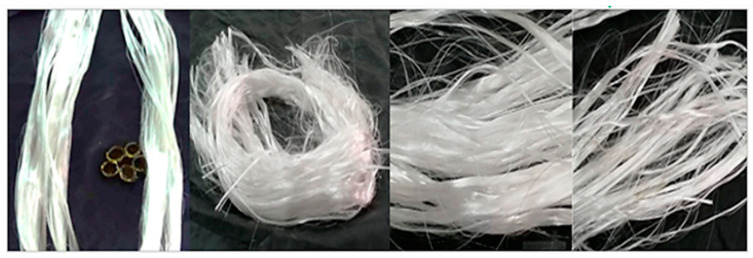
Produced Phosphate glass fibers.

**Figure 2 materials-15-07601-f002:**
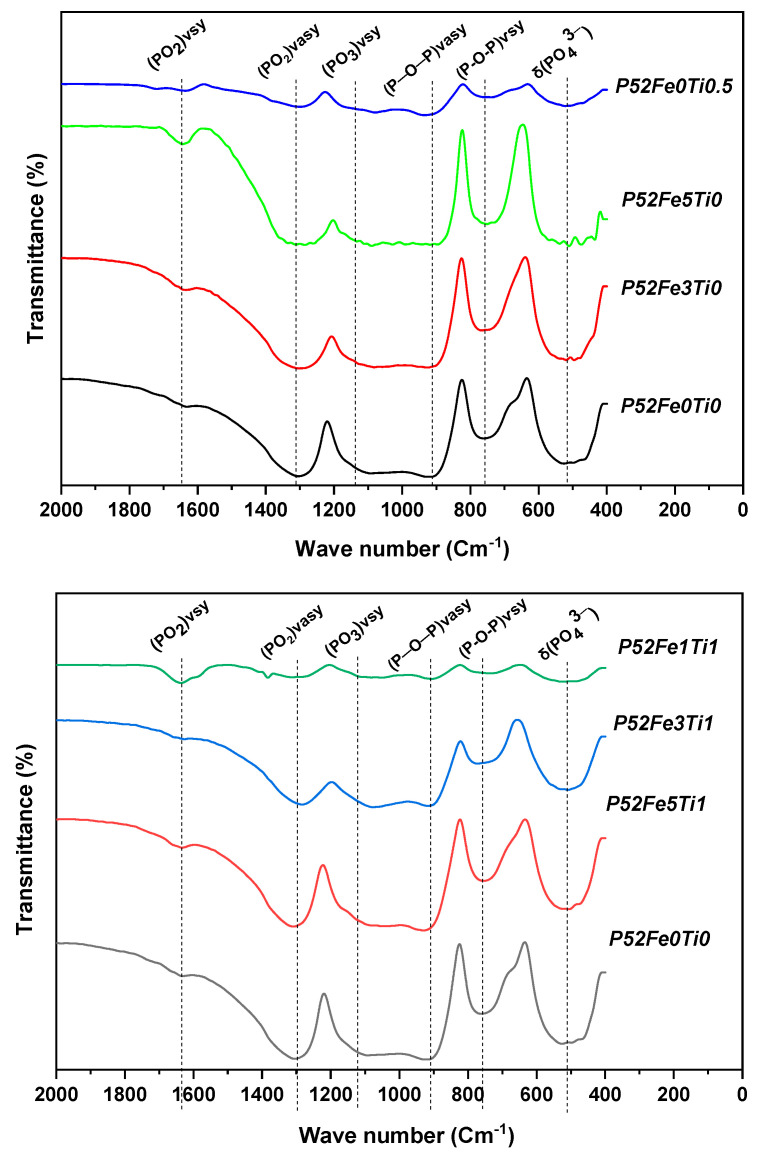
IR spectra of phosphate glasses with different compositions.

**Figure 3 materials-15-07601-f003:**
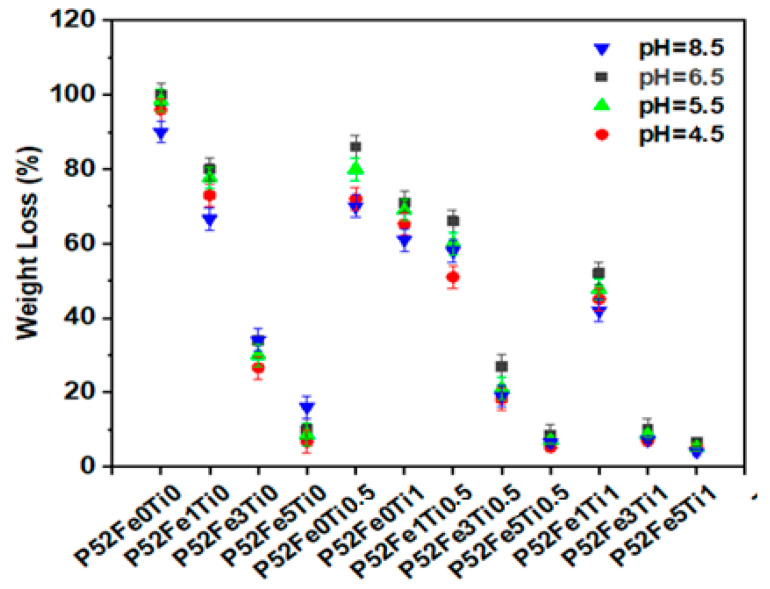
Weight loss Percentage of Phosphate glass fibers in an aqueous medium.

**Figure 4 materials-15-07601-f004:**
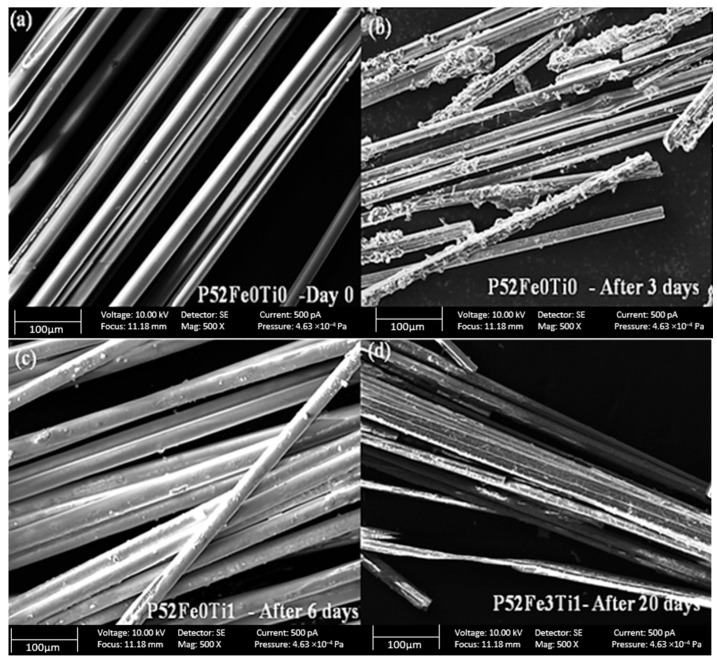
The morphology of PGFs without Fe_2_O_3_/TiO_2_ before (**a**), after (**b**) the degradation, and PGFs doped by Fe_2_O_3_/TiO_2_ after 6 days (**c**) and 20 days (**d**) of the degradation tests.

**Figure 5 materials-15-07601-f005:**
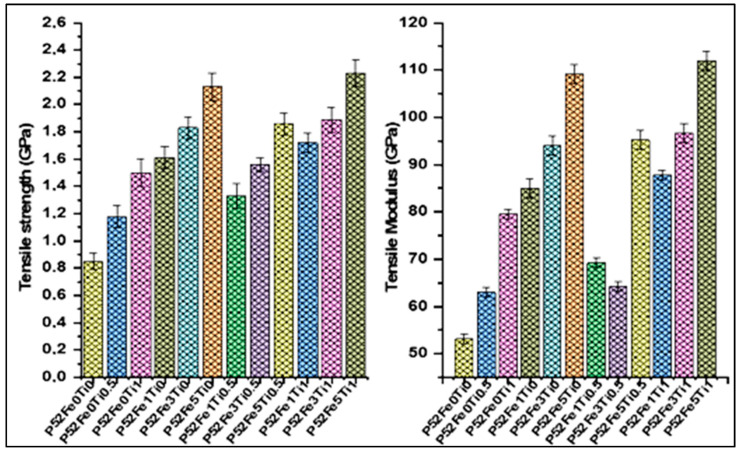
Tensile strength and tensile modulus of phosphate glass fibers.

**Figure 6 materials-15-07601-f006:**
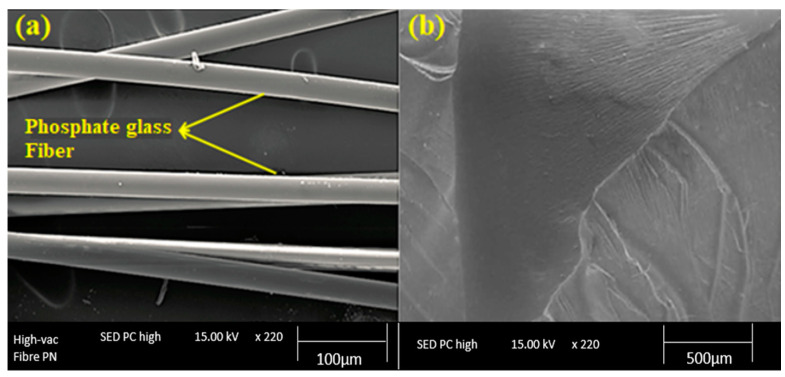
SEM images of the reinforcement (**a**) and the polyester matrix (**b**).

**Figure 7 materials-15-07601-f007:**
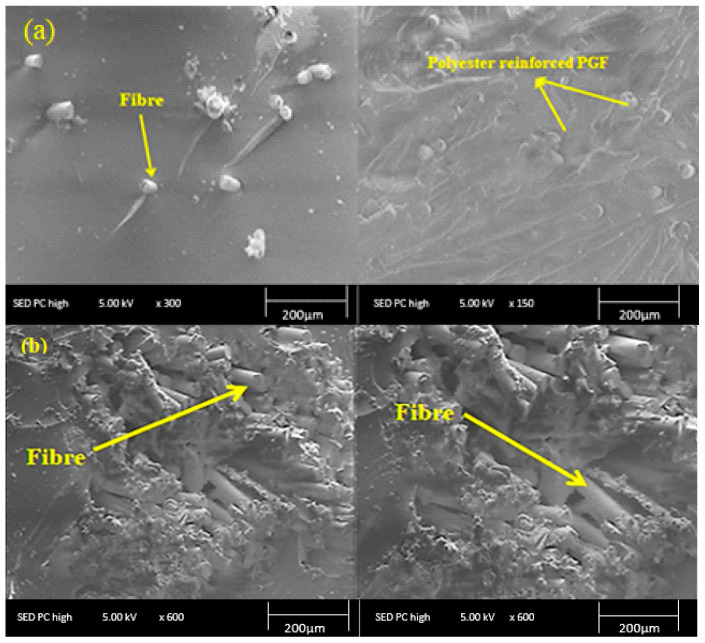
Morphology of polyester composites reinforced with 16 wt % of PGF (**a**) and 20 wt % of PGF (**b**).

**Figure 8 materials-15-07601-f008:**
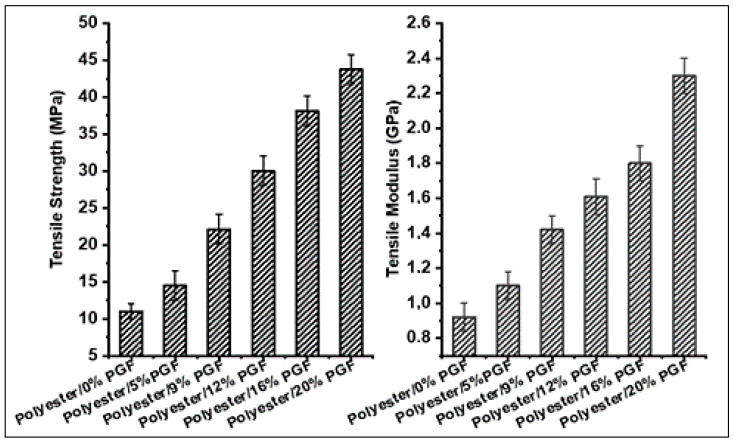
Effect of phosphate glass fiber content on the tensile strength and modulus.

**Figure 9 materials-15-07601-f009:**
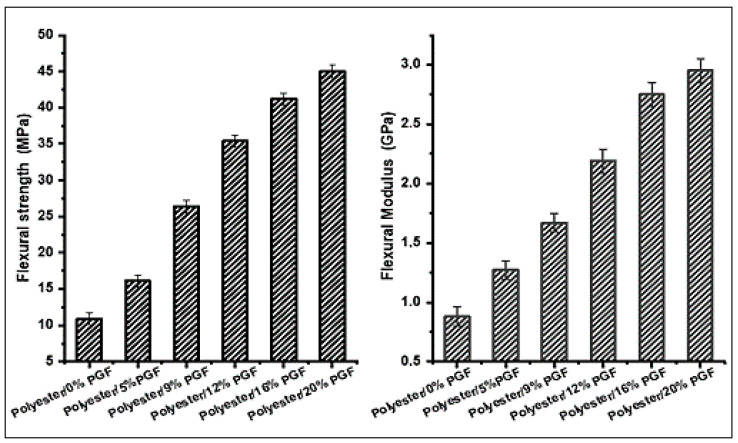
Effect of phosphate glass fiber content on flexural strength and modulus.

**Figure 10 materials-15-07601-f010:**
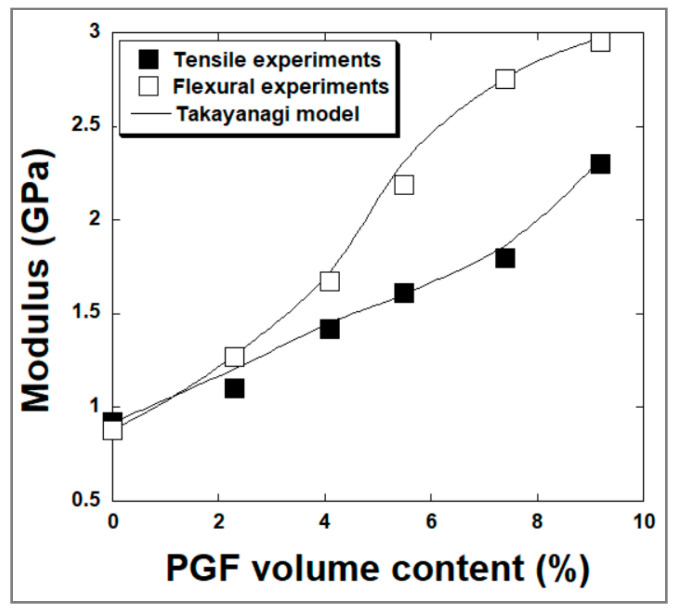
Fitting by Takayanagi model on the evolution of the modulus of sPGF/polyester-based composites vs. filler volume fraction.

**Table 1 materials-15-07601-t001:** Composition (%Mol) and density of phosphate glass fibers.

Glass Samples	P_2_O_5_	CaO	K_2_O	MgO	Fe_2_O_3_	TiO_2_	O/P	Density (g/cm^3^)
P52Fe0Ti0	52	24	11	13	0	0	2.96	2.37 ± 0.03
P52Fe0Ti0.5	52	24	10.5	13	0	0.5	2.96	2.42 ± 0.03
P52Fe0Ti1	52	24	10	13	0	1	2.97	2.46 ± 0.03
P52Fe1Ti0	52	24	10	13	1	0	2.98	2.79 ± 0.03
P52Fe3Ti0	52	24	8	13	3	0	3.01	2.48 ± 0.03
P52Fe5Ti0	52	24	6	13	5	0	3.05	2.56 ± 0.03
P52Fe1Ti0.5	52	24	9.5	13	1	0.5	2.98	2.46 ± 0.03
P52Fe3Ti0.5	52	24	7.5	13	3	0.5	3.02	2.52 ± 0.03
P52Fe5Ti0.5	52	24	5.5	13	5	0.5	3.06	2.69 ± 0.03
P52Fe1Ti1	52	24	9	13	1	1	2.99	2.56 ± 0.03
P52Fe3Ti1	52	24	7	13	3	1	3.02	2.63 ± 0.03
P52Fe5Ti1	52	24	5	13	5	1	3.06	2.79 ± 0.03

**Table 2 materials-15-07601-t002:** The dissolution rate of Phosphate glasses fibers.

PGF Samples	DR (g·cm^−2^·min^−1^) × 10^−15^			
	pH = 4.5	pH = 5.5	pH = 6.5	pH = 8.5
P52Fe0Ti0	4.65	7.62	11	3.76
P52Fe1Ti0	1.16	2.87	5.6	1.06
P52Fe3Ti0	0.42	0.58	1.3	0.33
P52Fe5Ti0	0.098	0.56	0.19	0.14
P52Fe0Ti0.5	2.8	5.26	7.5	2.93
P52Fe0Ti1	1.11	2.17	3	1.39
P52Fe1Ti0.5	0.83	0.96	2.1	0.88
P52Fe3Ti0.5	0.24	0.31	0.52	0.19
P52Fe5Ti0.5	0.048	0.076	0.0927	0.066
P52Fe1Ti1	0.53	0.53	1.09	0.35
P52Fe3Ti1	0.071	0.098	1.5	0.062
P52Fe5Ti1	0.049	0.051	0.0535	0.025

**Table 3 materials-15-07601-t003:** Characteristics of developed composite materials.

Samples	Mass Fraction (%)	Composite Density g/cm^3^	Volume Fraction (%)	Thickness (mm)
Polyester/5% PGF	5	1.24 ± 0.01	2.3	5
Polyester/9% PGF	9	1.26 ± 0.01	4.1	5
Polyester/12% PGF	12	1.30 ± 0.01	5.5	5
Polyester/16% PGF	16	1.34 ± 0.01	7.4	5
Polyester/20% PGF	20	1.36 ± 0.01	9.2	5

## Data Availability

The raw data cannot be shared at this time as the data are also part of an ongoing study.

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
