# Peer review of "Development and Characterization of Phosphate Glass Fibers and Their Application in the Reinforcement of Polyester Matrix Composites"

_materials, 2022, doi:10.3390/ma15217601_

Round 1
Reviewer 1 Report
Interesting work. Major revisions are in order for the authors to address the comments detailed in the points below:
Language needs to be significantly revised. Several minor mistakes found.
“many industrial and technological applications are interested in composite-based components as materials of the future [1] due to their outstanding properties such as high resistance to fatigue and corrosion”: see for example 10.1016/j.conbuildmat.2022.126320 and 10.3390/ma15041596 and further complement.
“matrix-based composites is the type of reinforcements used in terms of cost and their compatibility (physical and chemical) with the matri”: but why? Needs to be clarified.
“ave proved that TiO2 metal oxide is effective for the enhancement of PG physical properties”: what properties? Must be detailed.
“. The spinning temperature was programmed at 600 °C and the fibers were pu”: why this temperature? Detail.
Fig 1: scale is missing.
Why drives such a significant change in the weight loss between different conditions as shown in figure 2?
Fig 3: scales are not properly seen.
In fig 3 b what is the particles that adhered to the fibers?
“ddition of Fe2O3/TiO2 in the composition of phosphate glasses improves their mechanical properties and chemical durability”: but why is that?
How does the Young modulus compares to the literature?
“failure phenomenon was observed in the case of the polyester composite reinforced with glass fibers”: where is this shown?
Can the authors comment on the fibers fracture that is shown in fig 6?
“The tensile strengths of the polyester composites increased from (11±1) MPa to (22.12 ± 2) MPa with increasing fiber content from 0%vol to 4.1%vol and achieved its maximum value of (43.75±2) MPa with increasing fiber content to 9.2%vol.”: how about ductility? Any changes?
Please also show the tensile curves to see the evolution during loading.
“A Takayanagi model was used to approximate the moduli values of the composit”: why was this model used? Must be detailed and clarified.
Have the authors checked the fracture surfaces of the tensile specimens?
Author Response
We would like to thank the reviewers for the careful and thorough reading of our manuscript and for the thoughtful comments and constructive suggestions, which help to improve the quality of this manuscript.
Please see the attachment our responses to your remarks

Reviewer 2 Report
the title should be modified if the study include testing the composites as well - not just glass fibre - confusing title
why phosphate glass fibre different from ordinary glass fibre?
chemical durability - explain
methodology to characterize composites should separate from fabrication of composites
It is too short - write all methods used
Fig 2 - is this the results of chemical durability
section 3.2.3.1 avoid using results because section 3 is results and discussion not just results
3.2.3.1 should be supported by published work
Avoid et al in references
Author Response
We would like to thank the reviewers for the careful and thorough reading of our manuscript and for the thoughtful comments and constructive suggestions, which help to improve the quality of this manuscript.
Please see the attachment, our responses to your remarks

Reviewer 3 Report
This paper investigated the mechanical properties of phosphate glass fibers reinforced polyester composites. Some important information and results have been further obtained. However, to improve the quality of the paper, the following comments should be considered.
1. Abstract, please indicate the dispersion state of glass fiber in polyester systems, which is critical for improving the mechanical performance of resin. In addition, what is the improving mechanism the mechanical properties of the resin after adding fibers? In the beginning, the advantages of phosphate glass fiber should be further emphasized.
2. Introduction, the summary of the properties and advantages of two kinds of fiber reinforced polymer composites in paragraph one is not comprehensive. This paper focuses on glass fiber composites. It is suggested that the authors should summarize and compare the performance and price of CFRP and GFRP to emphasize the advantages of GFRP related to this paper. For example, CFRP has very excellent mechanical properties, corrosion and fatigue resistances. However, the higher price and the lower fracture toughness are the main factors limiting the engineering application. In contrast, GFRP has a lower price and better toughness, which is popular in some engineering applications. Please review the latest research below to further summarize the properties and advantages of this composites. International Journal of Fatigue, 2020, 134: 105480. Composite Structures. 2021, 261: 113285. Composite Structures, 2022, 293, 115719.
3. This paper focuses on short phosphate glass fibers reinforced polyester composites. However, reviewers did not see the summary on the mechanical properties and dispersion mechanism in the introduction. In addition, why do you use chopped fiber? It is well known that continuous fiber reinforced polymer composites have very excellent properties, which can maximize the performance of fibers. Please provide relevant explanations.
4. Is the synthesis and preparation of glass fiber in Section 2.1 carried out in the current laboratory? How are some preparation parameters determined?
5. Please provide some information about the source and manufacturer of polyester resin.
6. In part 2.2.2, the length of glass fiber is determined as 3mm. Is there any basis and source for determining this length? Too large or too small length is unfavorable for improving the performance of resin.
7. Please provide some specific details about the test methods of physical, mechanical and morphological properties of composite materials. For example, testing rate, sample number and loading method, etc.
8. Part 3.1.1, the chemical durability of glass fiber is currently characterized by the method of mass loss. However, this may be insufficient. In addition to mass loss, you should consider using some micro means, such as FTIR, XPS and other means to analyze the changes of some elements and functional groups.
9. It can be found in Figure 2 and Table 2 that PH values are basically acidic. Why not use alkaline pH value? Generally, the Si-O skeleton of glass fiber will react with hydroxide ions in alkali solution. This reaction may destroy the structure of glass fiber to a large extent.
10. In part 3.1.3, for the analysis of single fiber testing, Weibull distribution is generally used to obtain the shape parameters and scale parameters. This is also a very effective method to evaluate the dispersion of fiber strength.
11. The definition of picture 6 is very poor. It is recommended to further improve it.
12. In part 3.2.3, the mechanical properties of the composite materials have been obtained and analyzed. However, there is a lack of analysis of some relevant mechanisms, such as Polymers. 2022, 14, 1087. This information may be very important for the reader to understand how glass fibers improve resins. It is suggested that the authors supplement the mechanism analysis, for example, the dispersion state of fiber, the interface performance of fiber/resin, etc.
Author Response
We would like to thank the reviewers for the careful and thorough reading of our manuscript and for the thoughtful comments and constructive suggestions, which help to improve the quality of this manuscript.
Please see the attachment,our responses to your remarks

Reviewer 4 Report
The authors manufactured a series of phosphate glass fibers of different composition, and evaluate their mechanical properties and the applicability in polyester matrix composites. The methodology is clearly written and the conclusion is supported by the results. I have some clarification questions:
1. What is the mechanism of bonding between the fibers and the matrix, and what is the estimated interfacial strength?
2. For the measurement related to eq.2, what is the strain at which the composite modulus is measured? I expect no interfacial failure at small strain but debonding star to happen at larger strains. What is the strain level associate to the parameter \phi of value 0.97 to 0.98?
3. In addition to increasing the strength, how does the addition of fiber change the deformability of the matrix?
Author Response
We would like to thank the reviewer for the careful and thorough reading of our manuscript and for the thoughtful comments and constructive suggestions, which help to improve the quality of this manuscript.
Please see the attachment, Our responses to your remarks

Round 2
Reviewer 1 Report
Not all comments were addressed. See below:
Language needs to be significantly revised. Several minor mistakes found.
“many industrial and technological applications are interested in composite-based components as materials of the future [1] due to their outstanding properties such as high resistance to fatigue and corrosion”: see for example 10.1016/j.conbuildmat.2022.126320 and 10.3390/ma15041596 and further complement.
Why drives such a significant change in the weight loss between different conditions as shown in figure 2? Still unclear. Must be critically discussed.
Please also show the tensile curves to see the evolution during loading. Not shown.
Have the authors checked the fracture surfaces of the tensile specimens? Not shown.
Author Response
We would like to thank the reviewer for the careful and thorough reading of our manuscript and for the thoughtful comments and constructive suggestions, which help to improve the quality of this manuscript. Also, proofreading was done.
Please see in the attachment our responses to your remarks

Reviewer 3 Report
It can be accepted.
Author Response
We would like to thank the reviewer 3 for his careful and thorough reading of our manuscript and for his positive answer
Round 3
Reviewer 1 Report
Acceptance is recommended